# Psychological distress among parents with emigrant offspring: A mixed-methods study from Changunarayan Municipality, Nepal

Anjani Bhandari[1ʘ], Shishir Paudel[1,2ʘ], Anisha Chalise[3*]

1 Department of Public Health, CiST College, Pokhara University, Kathmandu, Nepal, 2 Kathmandu Institute of Child Health, Hepali Heights, Budhanilkantha, Kathmandu, Nepal, 3 Center for Research on Environment, Health and Population Activities (CREHPA), Lalitpur, Nepal

ʘ These authors contributed equally to this work.
* anisha.chalise90@gmail.com

## Abstract

### Background

The growing trend of international migration has significant socio-emotional implications for families left behind, particularly the left-behind parents. While much research focuses on the well-being of migrants, limited studies explore the psychological distress experienced by left-behind parents. This study assesses the prevalence and determinants of psychological distress among parents with emigrant offspring in Changunarayan Municipality, Nepal.

### Method

A mixed-methods cross-sectional study was conducted among 218 parents whose children had emigrated for at least six months. The quantitative phase involved a structured survey, where psychological distress was assessed using the Kessler Psychological Distress Scale (K10). Pearson chi-square tests and multivariable logistic regression were performed to identify associated factors at a 5% level of significance. The qualitative phase included 16 in-depth interviews to explore emotional experiences and coping strategies. Thematic analysis, following Braun and Clarke's six-step framework, was used to identify key qualitative insights.

### Results

The prevalence of psychological distress was 18.8%, with 8.3% experiencing mild distress, 5.0% moderate distress, and 5.5% severe distress. Multivariable analysis revealed that parents with multiple morbidities (aOR: 4.032, 95% CI: 1.633–9.938), those whose children were employed in labor-intensive jobs (aOR: 9.215, 95% CI: 1.499–56.645), and those perceiving low support from emigrant children (aOR: 3.828, 95% CI: 1.178–12.442) had significantly higher odds of psychological distress.

**Data availability statement:** All relevant data related to quantitative analysis is available within the manuscript and its supplementary information file as S2 File. All relevant qualitative data are within the paper.

**Funding:** The author(s) received no specific funding for this work.

**Competing interests:** The authors have declared that no competing interests exist.

Qualitative findings highlighted loneliness, parental worries, and uncertainty regarding children's return as key concerns. Social support, regular communication, and engagement in community activities were identified as coping strategies.

## Conclusion

A significant proportion of parents with emigrant children experience psychological distress, influenced by health conditions, migrant employment status, and perceived social support. Strengthening intergenerational communication, enhancing mental health services, and developing targeted support programs for left-behind parents are crucial in mitigating their distress.

---

## Introduction

International migration has increased significantly in recent decades, shaping economies, societies, and family structures worldwide [1,2]. According to the United Nations Department of Economic and Social Affairs (UN DESA), in 2020, approximately 281 million individuals were living as international migrants, comprising 3.6% of the global population [2]. Migration is primarily driven by economic opportunities, climate change, and political instability, affecting not only the migrants themselves but also their families left behind [3]. While much attention has been given to the mental health of migrants, less focus has been placed on the psychological well-being of those they leave behind, particularly left-behind parents. Migrants, particularly refugees and asylum seekers, are at high risk of experiencing mental health problems. Pre-migration trauma, violence, and persecution, combined with post-migration challenges such as discrimination, uncertain legal status, language barriers, and unemployment, contribute to conditions like depression, anxiety, and post-traumatic stress disorders (PTSD) not only among migrants but also among their left-behind family members [4–7].

In Nepal, labor migration is a well-established phenomenon, with over 5.9 million Nepalese employed overseas [8]. Recent estimates indicate that approximately 1,700 Nepalese leave the country daily in search of employment or education [9]. The Nepalese economy heavily relies on remittances, which contributed 25.33% to the national Gross Domestic Product (GDP) in 2024, amounting to $10.86 billion [8]. While remittances improve household income and living conditions [5,10], the migration of children can lead to significant emotional and social consequences for those parents who are left behind. Migration often results in disrupted caregiving structures, weakened intergenerational ties, and increased social isolation, which can contribute to psychological distress among left-behind parents [5–7].

Parental well-being is deeply influenced by the migration of their children, often resulting in emotional and psychological challenges. Studies indicate that left-behind parents frequently experience loneliness, social isolation, and a lack of essential support, which can exacerbate their risk of mental health disorders [11,12]. Global studies report varied prevalence rates of psychological distress among left-behind

parents [13–15]. For instance, research from neighboring country China reported a depression rate of 13% among parents of emigrant children [16]. Another study from neighboring country India found that 17% of left-behind parents experienced depressive symptoms [17]. In Sri Lanka, the prevalence of common mental disorders (CMD), including depression, somatoform disorder, and anxiety, was nearly 21% among left-behind parents [18]. The impact of child migration on the well-being of left-behind parents is a complex issue with conflicting research findings. While some studies indicate a correlation between child migration and negative mental health outcomes for parents, such as increased depression [13,19], loneliness [13,19,20], anxiety [19,21] and poorer cognitive ability [15], other research suggests that left-behind parents may have a lower risk of developing depression [22], higher cognitive function [23], or no significant differences [10,24]. In context of Nepal, studies have reported significantly high distress levels among left-behind parents, with anxiety affecting 56% and depression rates ranging between 26% and 36% [10,25].

In Nepalese society, intergenerational support is a cultural expectation [26]. However, with nearly 23% of Nepal's population currently residing abroad, traditional support systems might have been weakened, leaving the parents more vulnerable to mental health challenges [27–29]. This shift in family dynamics might have created an emotional ambivalence among parents, balancing pride in their children's achievements with the distress of physical and emotional separation [30]. While migration's economic impact has been widely studied, its psychological consequences on left-behind parents remain underexplored, particularly in Nepal. Existing research largely focuses on financial benefits, while neglecting the significant emotional and psychological burden experienced by aging parents. Additionally, there is limited understanding of how left-behind parents navigate emotional distress, where they seek support, and what coping mechanisms they employ [31]. This study aims to bridge that gap by assessing the prevalence and key determinants of psychological distress among parents of emigrant children, while also exploring coping strategies they adopt. The findings will provide valuable insights to inform healthcare policies and social welfare programs aimed at improving the mental well-being of left-behind parents in Nepal.

## Materials and methods

### Study design and setting

A cross-sectional study was conducted among the residents of Changunarayan Municipality in Bhaktapur District, Nepal. The study incorporated a quantitative survey followed by a qualitative phase to gather comprehensive insights. Changunarayan Municipality, located in Bagmati Province consists of approximately 21,588 households and 88,083 residents [9]. The municipality comprises nine wards, all of which were selected as the study setting. The target population consisted of residents whose children had emigrated for more than six months before data collection. There were no specific exclusion criteria. Participants completing the quantitative survey were eligible for inclusion in the qualitative phase.

### Sampling procedures and participants' recruitment

The sample for the quantitative phase was determined using Cochran's formula for the estimation of a proportion ($n = z^2pq/d^2$). A previous study assessing psychosocial problems in families of migrant workers across nine districts of Nepal reported worry symptoms in 10.8% and depression symptoms in 5.7% of mothers, yielding a combined prevalence of 16.5% [32]. Considering this past prevalence at 5% allowable error, and a 95% confidence interval, the required sample size was 212. Adjusting for a 5% non-response rate, the sample size was optimized to 223. A total of 223 participants were approached, and 218 complete responses were obtained, satisfying the required sample size and the quantitative phase was concluded. Due to the absence of a comprehensive record of households with emigrant children, purposive sampling technique was employed. Initially, in each ward, the screening for eligibility began from the first household adjacent to the ward office, and subsequent households were approached sequentially, until a total of 24–25 households provided their complete response. If both parents were present in an eligible household, male and female participants were alternated between households to ensure balanced representation. This process was followed in each of the nine wards.

 

For the qualitative phase, participants from the quantitative survey were purposively selected based on their willingness to participate and socioeconomic characteristics. The number of in-depth interviews (IDIs) was determined using the principle of data saturation, where interviews continued until no new themes emerged. Saturation was reached after 16 interviews (Table 1).

## Data collection

The data for both the quantitative and qualitative phases of this study were collected through face-to-face interviews. For the quantitative phase, data collection was conducted from June 10 2024 to July 30 2024 in participants' households using a structured questionnaire with close-ended questions. Informed consent was obtained from all participants before the interviews. The questionnaire covered four sections: socio-demographic and health related characteristics; emigration-related characteristics of children; perceived social support assessed through the modified Multidimensional Scale of Perceived Social Support (MSPSS) [33]; and psychological distress, assessed using the Kessler Psychological Distress Scale (K10) [34]. The questionnaire was translated into Nepali and back-translated into English to ensure linguistic accuracy and validity. Pretesting was conducted among 10% of the sample population in Bhaktapur Municipality. The Cronbach's alpha coefficients for the MSPSS and K10 scales were 0.831 and 0.851, respectively, indicating good internal consistency and reliability.

For the qualitative phase, an open-ended interview guide was employed to gather in-depth insights into participants' psychological well-being and coping strategies for distress related to their emigrant children. The open-ended interview guide was developed by AB under the guidance of SP and AC. This interview guide was piloted with three parents of emigrant children residing in Bhaktapur Municipality, and revisions were made based on their feedback. A total of 16 IDIs, each lasting 30–45 minutes, were conducted in private settings at participants' households to facilitate open discussions. All interviews were audio-recorded with participants' consent. AB was trained by senior researchers (SP and AC) in ethical

**Table 1. Demographic and family characteristics of IDI participants.**

| IDI no. | Age | Gender of Participants | Marital status of Participants | Education of Participants | Total no. of Children of Participants | No. of Emigrant Children | Country of Emigration |
|---|---|---|---|---|---|---|---|
| 1 | 59 | Male | Married | 12 | 4 | 1 | Australia |
| 2 | 36 | Female | Married | 10 | 2 | 1 | UAE |
| 3 | 52 | Female | Married | 5 | 2 | 1 | UAE |
| 4 | 55 | Male | Married | 12 | 2 | 1 | Australia |
| 5 | 57 | Female | Married | 10 | 2 | 1 | Canada |
| 6 | 63 | Female | Widow | 9 | 3 | 1 | UK |
| 7 | 69 | Male | Married | Bachelors | 3 | 1 | USA |
| 8 | 52 | Female | Married | 12 | 1 | 1 | USA |
| 9 | 75 | Male | Married | 10 | 2 | 2 | Australia & USA |
| 10 | 61 | Female | Widow | 8 | 1 | 1 | USA |
| 11 | 58 | Female | Married | 5 | 3 | 1 | Cyprus |
| 12 | 66 | Female | Widow | Illiterate | 3 | 1 | Malaysia |
| 13 | 71 | Male | Widower | 5 | 4 | 2 | UAE & Japan |
| 14 | 66 | Female | Married | 9 | 3 | 1 | Japan |
| 15 | 58 | Male | Married | 10 | 2 | 2 | Kuwait & Malaysia |
| 16 | 48 | Female | Married | Bachelor | 2 | 1 | UAE |

considerations, rapport building, maintaining participant confidentiality, managing emotional distress during interviews, and qualitative interview techniques.

## Study variables

Psychological distress was the outcome variable, assessed using the K10 scale, with scores interpreted as 10–19 (low/no distress), 20–24 (mild distress), 25–29 (moderate distress), and 30–50 (severe distress) [34]. For analysis, mild, moderate, and severe distress categories were categorized as the presence of psychological distress, while low/no distress was classified as the absence of psychological distress. The independent variables included socio-demographic and health related factors such as age, gender, education, occupation, marital status, the number of family members living together, and the presence or absence of chronic conditions. Emigration-related characteristics of children included the number of migrant children, duration of migration, frequency of communication, and financial support from emigrant children. Perceived social support was assessed using the modified MSPSS, where questions related to family support were rephrased to specifically reflect support from emigrant children. The MSPSS consists of three sub-scales measuring support from significant others, family, and support from friends. Scores were categorized as low support (1–2.9), moderate support (3–5), or high support (5.1–7) [34]. For further analysis, moderate and high support was grouped together.

## Data management and analysis

The data from the quantitative survey were entered using EpiData version 3.1 and exported to Statistical Package for Social Sciences version 22 for analysis. Descriptive statistics were used to summarize participants' demographic profiles, lifestyle, health-related characteristics, emigration characteristics, perceived social support, and level of psychological distress. Pearson chi-square test was performed at a 5% level of significance to identify the factors associated with psychological distress among the parents. Those factors found to have statistical significance (p < 0.05) in the chi-square test were subjected to the final model of multivariable logistic regression analysis. Prior to the multivariable analysis, the multi-collinearity among independent variables was tested using the Variance Inflation Factor (VIF), where a VIF greater than five was taken as an indication of multi-collinearity between the independent variables. Hosmer and Lemeshow test were used to assess the goodness of fit of the model where p < 0.05 represents that the model is a poor fit.

Qualitative data from IDIs were transcribed in Nepali and translated into English by AB. Transcripts were cross-checked for accuracy and consistency by SP and AC. Prolonged engagement of researchers in every phase of the study and peer debriefing for translations of verbatim were done. Due to the unavailability of all the participants in their households, the transcripts were not sent to the participants for their review. Braun and Clark's six-step thematic analysis with an inductive-deductive continuum was used for qualitative data analysis [35]. The interview transcripts and notes were read and analyzed several times to understand the emotions and experiences of participants and to become thoroughly familiar with the content. The qualitative data was analyzed manually, without using any data coding and analysis software. Based on the information provided by the participants, the core concepts expressed by each of the participants were extracted as codes and further clustered into themes. All authors collaboratively reviewed and finalized the themes. The initial codes were prepared by the first author, who is a public health undergraduate. The codes were reviewed and clustered into themes by AC and SP, who are public health postgraduates with past experiences in both quantitative and qualitative data analyses.

## Ethical consideration

The ethical approval for this study was obtained from the Institutional Review Committee of CiST College (Registration no: 35/080/081). Written informed consent, in the form of signature or thumbprint was obtained from all the participants before data collection in both the quantitative and qualitative phases. The participants of the IDIs were also asked for their permission to audio-record the interview sessions. Participant privacy and confidentiality were maintained throughout, and all identifying information was anonymized using participant codes.

## Results

### Quantitative results

Out of the total of 218 participants, majority (81.2%, 95% CI: 75.7–86.0) were psychologically sound while 41(18.8%) were found to have psychological distress. Based on the Kessler Psychological Distress Scale, 8.3% of the participants had mild distress, 5.0% had moderate distress, and 5.5% experienced severe distress (Table 2).

Among 218 participants, nearly two-thirds (63.3%) were females, while 36.7% were males. The age of the participants ranged from 30 to 82 years, with the mean age of 55.15±9.733 years. In context of education level of the participants, more than a fourth of them (27.1%) had completed secondary level education, closely followed by those who had informal education (26.1%). More than two-fifths of the participants (44.5%) had at least one chronic condition, with 22.5% having a single morbidity and 22.0% experiencing multimorbidity. The most commonly diagnosed conditions included hypertension (n=57), diabetes (n=35), arthritis (n=7), asthma (n=7), and chronic kidney disease (n=3). Additionally, 47 participants reported other diagnosed chronic conditions, among which high cholesterol (n=11) and cardiovascular diseases (n=8) were the most frequent. There was no statistically significant relationship observed between psychological distress and variables such as age, sex, religion, ethnicity, number of family members, educational status, and occupation. The variables statistically associated with psychological distress were marital status of participants (p<0.05) and chronic conditions (p=0.007), (Table 3).

Most of the participants perceived receiving moderate to high support from their significant other (82.6%), emigrant children (92.7%) and friends (56.0%). Psychological distress was significantly associated with perceived support from significant others (p=0.007), and perceived support from migrant children (p=0.008) (Table 4).

Regarding the emigrant children, more than three-quarters (83.0%) of participants had a single child living abroad. The average age of emigrants was 29.89±6.34 years, ranging from 18 to 53 years. The proportion of emigrant males (67.9%) was higher than emigrant females (32.1%). More than three-quarters (78.4%) had migrated for a longer term (≥2 years), while 21.6% were recent migrants (<1 year). Emigrants were distributed across several countries, including Australia (31.7%), Gulf countries (30.3%), the USA/Canada (22.5%), and the UK/Europe (15.6%). Nearly half of the emigrants (47.2%) held a working or dependent visa, followed by student visas (30.3%), while 22.5% were permanent residents. Nearly half of the emigrants (49.5%) communicated with their families on a daily basis while others communicated weekly (44.0%), and few communicated monthly (6.4%). A significant proportion of emigrants (69.7%) provided financial support to their families, while the remaining (30.3%) did not. Most of the emigrants visited home frequently (67.4%). Among the characteristics analyzed, only the occupation of the migrant was found to be significantly associated with psychological distress (p=0.023) (Table 5).

For multivariable analysis, the VIF test was performed among independent variables found to have statistically significant relationship with psychological distress in bivariate analysis, where the highest reported VIF was 1.598, indicating that

**Table 2. Psychological distress among the parents (n=218).**

| Variables | n (%) | 95% CI for Proportion |
|---|---|---|
| **Psychological Distress based on Kessler Score** | | |
| Psychologically Sound | 177 (81.2) | 75.7–86.0 |
| Mild Distress | 18 (8.3) | 5.0–12.8 |
| Moderate Distress | 11 (5.0) | 2.5–8.9 |
| Severe Distress | 12 (5.5) | 2.9–9.5 |
| **Psychological Distress Category** | | |
| Absence | 177 (81.2) | 75.7–86.0 |
| Presence | 41 (18.8) | 14.0–24.3 |

**Table 3. Sociodemographic and health related characteristics of the participants and psychological distress (n = 218).**

| Characteristics | n(%) | Psychological Distress | | p-value |
|---|---|---|---|---|
| | | Yes (n%) | No (n %) | |
| **Age** | | | | 0.455 |
| 30-40 years | 6(2.8) | 1(16.7) | 5(83.3) | |
| 41-50 years | 76(34.9) | 12(15.8) | 61(84.2) | |
| 51-60 years | 77(35.3) | 14(18.2) | 63(81.8) | |
| >60 years | 59(27.1) | 14(23.7) | 45(76.3) | |
| **Gender** | | | | 0.700 |
| Male | 80(36.7) | 10(12.5) | 70(87.5) | |
| Female | 138(63.3) | 31(22.5) | 107(77.5) | |
| **Ethnicity** | | | | 0.176 |
| Dalit | 3(1.4) | 2(66.7) | 1(33.3) | |
| Disadvantaged janajati | 16(7.3) | 2(12.5) | 14(87.5) | |
| Relatively advantaged janajati | 36(16.5) | 2(12.5) | 14(87.5) | |
| Brahmin/ Chhetri | 163(74.8) | 30(18.4) | 133(81.6) | |
| **Education Level** | | | | 0.268 |
| Illiterate | 23(10.6) | 7(30.4) | 16(69.6) | |
| Literate through informal education | 57(26.1) | 14(24.6) | 43(75.4) | |
| Primary Level (1–8) | 43(19.7) | 10(23.3) | 33(76.7) | |
| Secondary Level (9–10) | 59(27.1) | 6(10.2) | 53(89.8) | |
| Higher Secondary Level (11–12) | 20(9.2) | 4(20.0) | 16(80.0) | |
| Bachelor and higher degree | 16(7.3) | 0(0.0) | 16(100.0) | |
| **Marital Status** | | | | 0.014* |
| Married | 190 (87.2) | 31(16.3) | 159(83.7) | |
| Widow/widower | 28 (12.8) | 10(35.7) | 18(64.3) | |
| **Occupation** | | | | 0.215 |
| Homemaker | 76(34.9) | 18(23.7) | 58(76.3) | |
| Agriculture | 53(24.3) | 12(22.6) | 41(77.4) | |
| Retired | 42(19.3) | 3(7.1) | 39(92.9) | |
| Business | 38(17.4) | 7(18.4) | 31(81.6) | |
| Service | 9(4.1) | 1(11.1) | 8(88.9) | |
| **Chronic Conditions** | | | | 0.007* |
| No morbidity | 121(55.5) | 15(12.4) | 106(87.6) | |
| Single morbidity | 49(22.5) | 10(20.4) | 39(79.6) | |
| Multi-morbidity | 48(22.0) | 16(33.3) | 32(66.7) | |

*statistically significant at p < 0.05.

there was no issue of multicollinearity. In the multivariable analysis, participants suffering from multiple morbidities were found to have a four-fold increase in their odds of experiencing psychological distress (aOR: 4.032; 95% CI: 1.633–9.938) as compared to those with no morbidity. Additionally, participants whose children were employed as laborers had significantly higher odds of psychological distress (aOR: 9.215; 95% CI: 1.499–56.645) than those whose children were students. The parents who reported low level of perceived support from their migrant children were observed to have thrice the odds of experiencing psychological distress (aOR: 3.828; 95% CI: 1.178–12.442), in comparison to those who reported moderate to high level of perceived support from their migrant child. While experiencing lower level of perceived support from the significant other was associated with higher unadjusted odds of psychological distress (uOR: 2.823; 95%

**Table 4. Social support factors and psychological distress (n = 218).**

| Characteristics | n(%) | Psychological Distress | | p-value |
| --- | --- | --- | --- | --- |
| | | Yes (n%) | No (n%) | |
| **Support from Significant Other** | | | | |
| Low Support | 38(17.4) | 13(34.2) | 25(65.8) | 0.007* |
| Moderate/High Support | 180(82.6) | 28(15.6) | 152(84.4) | |
| **Support from Migrant Child** | | | | |
| Low Support | 16(7.3) | 7(43.8) | 9(56.3) | 0.008* |
| Moderate/High Support | 202(92.7) | 34(16.8) | 168(83.2) | |
| **Support from Friends** | | | | |
| Low Support | 96(44.0) | 23(24.0) | 73(76.0) | 0.084 |
| Moderate/High Support | 122(56.0) | 18(14.8) | 104(85.2) | |

*statistically significant at p < 0.05.

CI: 1.291–6.171; p = 0.009), this relationship became non-significant in the adjusted model (aOR: 1.945; 95% CI: 0.437–8.651) (Table 6).

The additional analysis treating psychological distress score as a continuous variable is provided in S1 and S2 Tables.

## Qualitative results

**Persistent emotional distress and loneliness.** Most of the parents expressed feelings of persistent loneliness, sadness, and emotional suffering due to the prolonged physical absence of their children. They shared that their everyday lives feel empty, and special occasions like festivals seemed less joyful without the presence of their children. Many also worried about growing old without their children by their side. The emotional pain of being separated was a recurring struggle for almost everyone. Participants also highlighted how this absence created a void in their daily routines and family dynamics. Their regular activities, such as eating meals or celebrating festivals, felt incomplete without their children. Beyond seasonal emotional distress, some parents also experienced deepening worry about long-term emotional well-being, questioning their role in their children's lives.

> *"We often get feelings of loneliness as we have been living on our own and in the future, we fear ending up with only the two of us."* **IDI 1.**

> *"I feel as though my home has become empty without my children. No matter how much I try to distract myself, there's always that quietness that reminds me they're not here."* **IDI 3.**

> *"Dashain and Tihar don't feel like festivals anymore. There's no joy in preparing food when they're not here to eat it with us."* **IDI 6.**

> *"What will happen to us when we are too weak to care for ourselves? They're so far away; I wonder if they'll even know if something happens to us."* **IDI 7.**

**Heightened anxiety over children's well-being.** The parents are concerned about their children's well-being abroad and their inability to provide immediate support. They wonder if their kids are eating well, staying healthy, or hiding their struggles to avoid causing worry back home.

> *"I constantly worry about my son. Is he eating properly? Is he getting enough rest? He doesn't always tell me when something is wrong because he doesn't want me to worry, but that makes me worry even more."* **IDI 9.**

**Table 5. Emigrants related characteristics and its association with psychological distress.**

| Characteristics | n(%) | Psychological Distress | | p-value |
|---|---|---|---|---|
| | | Yes (n%) | No (n%) | |
| **Number of Emigrant Children** | | | | 0.346 |
| Single child | 181(83.0) | 32(17.7) | 149(82.3) | |
| More than one child | 37(17.0) | 9(25.7) | 26(74.3) | |
| **Age of Emigrants** | | | | 0.528 |
| <25 years | 61(28.0) | 9(14.8) | 52(85.2) | |
| 25-30 years | 71(32.6) | 13(18.3) | 58(81.7) | |
| >30 years | 86(39.4) | 19(22.1) | 67(77.9) | |
| **Sex of Emigrants** | | | | 0.951 |
| Male | 148(67.9) | 28(18.9) | 120(81.1) | |
| Female | 70(32.1) | 13(18.6) | 57(81.4) | |
| **Duration of Migration** | | | | 0.183 |
| Recent migration (1 year) | 47(21.6) | 12(25.5) | 35(74.5) | |
| Long-term migration (≥2 years) | 171(78.4) | 29(17.0) | 142(83.0) | |
| **Country of Migration** | | | | 0.299 |
| Australia | 69(31.7) | 9(13.0) | 60(87.0) | |
| Golf countries | 66(30.3) | 15(22.7) | 51(77.3) | |
| USA/Canada | 49(22.5) | 8(16.3) | 41(83.7) | |
| UK/Europe | 34(15.6) | 9(26.5) | 25(73.5) | |
| **Nature of Visa** | | | | |
| Working visa/ dependent visa | 103(47.2) | 24(23.3) | 79(76.7) | 0.273 |
| Student visa | 66(30.3) | 10(15.2) | 56(84.8) | |
| Permanent resident | 49(22.5) | 7(14.3) | 42(85.7) | |
| **Type of Visa** | | | | 0.358 |
| Temporary | 169(77.5) | 34(20.1) | 135(79.9) | |
| Permanent | 49(22.5) | 7(14.3) | 42(85.7) | |
| **Occupation of Emigrant** | | | | 0.023* |
| Jobs | 164(75.2) | 27(16.5) | 137(83.5) | |
| Student | 47(21.6) | 10(21.3) | 37(78.7) | |
| Labor | 7(3.2) | 4(57.1) | 3(42.9) | |
| **Financial Support from Emigrant** | | | | |
| Yes | 152(69.7) | 24(15.8) | 128(84.2) | 0.084 |
| No | 66(30.3) | 17(25.8) | 49(74.2) | |
| **Frequency of Communication with Emigrant** | | | | |
| Daily | 108(49.5) | 18(16.7) | 90(83.3) | 0.576 |
| Weekly | 96(44.0) | 21(21.9) | 75(78.1) | |
| Monthly | 14(6.4) | 2(14.3) | 12(85.7) | |
| **Frequency of Home Visit** | | | | |
| Frequent (in every 1–2 years) | 147(67.4) | 32(21.8) | 115(78.2) | 0.107 |
| Infrequent | 71(32.6) | 9(12.7) | 62(87.3) | |

*statistically significant at p<0.05.

**Table 6. Factors associated with psychological distress.**

| Factors | uOR | p-value | aOR | p-value |
|---|---|---|---|---|
| **Marital Status** | | | | |
| Married | Ref | | | |
| Widow/widower | 2.849 (1.202-6.757) | 0.017* | 1.706 (0.316-9.208) | 0.535 |
| **Occupation of Emigrants** | | | | |
| Student | Ref | | | |
| Labor | 4.933 (0.946-25.737) | 0.058 | 9.215 (1.499-56.645) | 0.017* |
| Jobs | 0.729 (0.324-1.641) | 0.446 | 0.842 (0.335-2.118) | 0.715 |
| **Financial Support from Emigrant Children** | | | | |
| Yes | Ref | | | |
| No | 1.850 (0.916-3.738) | 0.086 | 1.946 (0.856-4.421) | 0.112 |
| **Chronic Conditions** | | | | |
| No morbidity | Ref | | | |
| Single morbidity | 1.812 (0.751-4.370) | 0.186 | 2.180 (0.844-5.632) | 0.108 |
| Multi-morbidity | 3.5333 (1.575-7.925) | 0.002* | 4.032 (1.633-9.938) | 0.002* |
| **Perceived Social Support from Emigrant Children** | | | | |
| Low Support | 3.843 (1.339-11.029) | 0.012* | 3.828 (1.178-12.442) | 0.026* |
| Moderate/High Support | Ref | | | |
| **Perceived Social Support from a Significant Other** | | | | |
| Low Support | 2.823 (1.291-6.171) | 0.009* | 1.945 (0.437-8.651) | 0.383 |
| Moderate/High Support | Ref | | | |

*statistically significant at $p < 0.05$.

*"A few weeks ago, my son fell sick, there is no one there to offer even hot water to him. If he was here with me (mother), I would have taken good care of him. He cares not to share unpleasant things to me which makes me feel bad."* **IDI 2.**

Financial sacrifices also weigh on many parents, as they've spent much of their savings to support their children's move abroad. These concerns are further amplified by global uncertainties or unforeseen challenges abroad, such as health issues or political instability, leading to emotional distress and self-doubt about whether the trade-off was worth their own mental suffering. For parents whose children were in physically demanding or unstable jobs, distress was compounded by worries over harsh labor conditions.

*"We sold a piece of land to send our daughter abroad, thinking she'd have a better life. Now, we have to live carefully with what we have, but I don't regret it because it's for her future."* **IDI 4.**

*Watching the news about the war in Israel gives me unpleasant thoughts for days. My daughter reassures me that she is safe and there's nothing to worry about where she lives. However, as a mother, I cannot help but worry about her."* **IDI 11.**

*"When I learned that my son has to stand for 9 hours as a security guard, I felt deeply saddened knowing how hard he works for his family."* **IDI 12.**

**Health challenges and emotional vulnerability.** Parents expressed concerns about their own physical and mental health in the absence of their children. Aging, chronic illnesses, and the lack of immediate family support amplified parents' psychological distress, as they faced fears of illness and dependency without their children's presence. Parents felt emotionally isolated, often suppressing their health concerns to avoid burdening their children.

*"I don't want to burden my children with my health issues. They have their own lives to handle abroad. So when we talk I keep things as normal as possible and don't tell them about our problems. But, honestly, sometimes it's hard. I wish them to be with me during my hard times."* **IDI 9.**

However, some parents expressed receiving some level of support, such as caretakers arranged by their children, which partially alleviated distress but did not eliminate feelings of loneliness and psychological vulnerability.

*"I have been taking medicines for thyroid, blood pressure, and diabetes for over 7 years. My children have arranged a caretaker who looks after my diet and checkups, constantly reassuring me that I'm not a burden and that my children still love me despite being far away."* **IDI 10.**

**Coping mechanisms as response to distress.**  Despite persistent distress, parents employed various strategies to manage their emotional struggles. Staying in regular contact with their children through calls or video chats has helped them feel connected. Some keep themselves busy with hobbies like gardening or helping out in their community. Many find comfort in talking to friends and neighbors in similar situations. Faith and spiritual practices also provide comfort, as parents pray for their children's safety and well-being. These little strategies help parents deal with the sadness of being apart.

*"I call them every day, sometimes twice. It's the only way I feel connected to their lives. Even hearing their voice for a few minutes makes my day."* **IDI 2.**

*"Since my daughter left for further studies, I miss the little things about her. We communicate at least twice a day. She shares everything with me about her day at college and work and I try to cheer her every time so she can focus on the goal she has dreamed of. I pray every day for their safety and happiness. It gives me strength to believe that one day, we'll all be together again."* **IDI 8.**

*"I've taken up gardening again. It helps me keep my mind off things, and it's nice to see something grow, even if my children are not here to see it."* **IDI 10.**

*"My neighbors have been a blessing. We often sit and talk about our children and share our worries. It helps to know I'm not the only one feeling this way."* **IDI 1.**

**Emotional conflict between pride and psychological distress.**  Parents expressed mixed feelings about their children's decision to migrate. On one hand, they're proud of their children for chasing better opportunities and building successful lives. On the other hand, they miss them deeply and wish they could come back home. A mix of pride and longing was observed among many parents as they struggle to balance their emotions.

*"I miss her every day, but I also know she's doing what's best for her career and her future. I'm proud of her, but my heart aches for her to be home."* **IDI 5.**

For some, broken promises of return exacerbated feelings of emotional distress, as their children gradually abandoned thoughts of coming back.

*"My son said that he will return back to Nepal once he completes his studies, a long time ago. Now he has no plan to return here. Instead, he requests us to move abroad with them. I know they can't return right now. They've built a life there, and I don't want to stand in the way of their success. But I hope they'll come back someday."* **IDI 7.**

*"I am quite hopeful they will return back and raise their kids with us. Me, and my wife can have our grandkids running around us soon."* **IDI 1.**

Others had accepted the reality of their children settling abroad permanently, but this did not lessen their emotional distress.

*"My son and his family are living a kind of luxury life abroad, which our country lacks in a lot of ways. If I ask my son to leave everything and come back to us, that won't be right. Our system does not have much opportunities to offer to them, which they already have abroad."* **IDI 4.**

*"Life in another country felt like a different world. I enjoyed my visit there, and I'm happy my son and daughter-in-law gave me the chance to experience it. The decision to return to Nepal should be theirs, they have struggled so much there, and now they finally have time to live in peace. I don't have anything to say about them returning."* **IDI 3.**

**Social perception and hidden emotional burden.** Participants described how societal perceptions added to their emotional burden. Some of the patents also shared that people assume they are well-off just because their children are abroad, but this isn't always true. Some parents also feel pressure to send their children abroad because it has become a common expectation in the community.

*"People think we are rich because our children are abroad, but they don't understand the sacrifices we've made. It's not easy living without them, and it's not easy managing everything on our own."* **IDI 11.**

*"All the young generation are leaving the country, and this has become normal in our society. I see everyone from my community has emigrated abroad for studies. My daughter also got influenced quickly and said she has no intention to return and settle here in Nepal."* **IDI 6.**

**Sources of happiness and hope.** Despite the emotional challenges, parents found reasons to remain hopeful and happy. Hearing about their children's achievements or successes gave them a sense of pride, making the sacrifices feel worthwhile. Many parents found strength in their faith and optimism, praying for their children's safety and hoping for a reunion in the future. They also drew comfort and courage from the support of their spouse or other family members, helping them endure the separation.

*"Though my elder son is away from home, I have my two younger sons and the eldest daughter-in-law who are here to support me. When I hear about my son's success at work in a foreign land, it makes me feel like all our struggles were worth it. I tell myself, we did the right thing by letting him go."* **IDI 12.**

*"Some days are harder than others, but knowing I have my loving wife who always supports me, gives me courage. We talk to our son every day, and sometimes it feels like he is just in another room."* **IDI 1.**

*"I pray every day for their safety and happiness. It gives me strength to believe that one day, we will all be together again."* **IDI 8.**

*"When my children tell me about their achievements, it makes me so happy. I know we made sacrifices, but they were worth it."* **IDI 4.**

This study employed a mixed-methods approach to explore psychological distress among parents of emigrants. The triangulation of findings reveals notable areas of convergence and divergence, enriching our understanding of the factors influencing psychological distress. While quantitative data identified significant predictors, qualitative narratives provided depth and context, highlighting the emotional and social dimensions of these experiences (Table 7).

**Table 7. Triangulation matrix for quantitative and qualitative findings.**

| Theme | Quantitative findings | Qualitative findings | Convergence/ Divergence |
|---|---|---|---|
| Prevalence of psychological distress | Psychological distress reported among 18.8% of the parents | Emotional distress and loneliness are recurring themes in IDIs | Convergence |
| Social Support | Perceived lower support from migrant children increased distress (aOR = 3.797). | Communication and reassurance mitigate distress. | Convergence |
| Impact of Chronic Conditions | Multi-morbidity increases distress (aOR = 4.168). | Chronic illness without support heightens distress. | Convergence |
| Parental Worries | Frequent communication indirectly mitigates distress. | Concerns about children's well-being abroad. | Convergence |
| Support from Significant Others | Initially significant but non-significant after adjustment | The support from the spouse were also a recurring coping measure suggested. | Divergence |
| Frequency of Communication | No significant differences in distress across communication frequency. | Frequent communication is described as essential. | Divergence |

## Discussion

This study explored the prevalence of psychological distress among parents with emigrant children and also identified factors associated with it, along with coping mechanisms. Psychological distress was observed among nearly a fifth of the left-behind parents. This prevalence is lower than the rate of depressive symptom among left-behind ageing parents in Krishnapur Municipality, Nepal, which was reported to be 35% [10]. The qualitative findings revealed a complex emotional landscape, where some parents reported distress and loneliness, while others expressed pride in their children's achievements and in some cases, the same parents experienced these ambivalent emotions simultaneously. The finding from this study in line with the studies from China [16], Moldova [36], and Thailand [22] where parents reported lower distress due to financial and social support from remaining family members and happiness for their migrant children as they were having better opportunities. In contrast, a study from neighboring country India found that older parents with all children living abroad experienced a higher prevalence of depression (24%) [37]. This may be due to the lack of support at home, leading to poor physical and mental health, and persistent worry about their children living far away. The variability in the rate of psychological distress across studies may be influenced by differing levels of financial support, communication frequency, and access to social resources.

In both qualitative and quantitative phases, psychological distress was expressed by the left-behind parents of emigrant children, but the study did not capture the mental health status of parents without emigrant children. However, several studies reveal contradicting findings regarding migration status of children and parental mental health. A study conducted in Nepal showed that the elderly parents having a migrant child had higher odds of self-perceived loneliness, while the odds of depressive symptoms, although higher, was not statistically significant [10]. Similarly, a study conducted in China, revealed that the mean level of loneliness significantly differed between empty nest older adults and non-empty nest older adults, and showed that empty nest older adults had significantly higher depression scores [13]. A study among Indian older adults revealed that higher proportion of the left-behind older adults showed signs of depression, compared to those who were not left-behind [17]. On the contrary, a study conducted in Thailand showed that the parents whose all children moved away were less likely to become depressed compared to those who had some or all children still living nearby, because migrant children often provided financial support which helps the parents, and the parents may worry less or feel less burdened when children are working and independent [22]. Given these mixed findings across different contexts, future research is needed to explore the relationship between migration of children and the mental health of older parents, particularly through longitudinal and comparative studies that account for cultural, economic, and support system differences.

In this study, the parents who were widow/widower were found to be twice more at odds of experiencing psychological distress in the unadjusted model (uOR: 2.849, p = 0.017). However, this association weakened and become non-significant in the adjusted model (aOR: 1.706, p = 0.535). Global studies present mixed findings, with some reporting widowhood as a risk factor for distress [38,39], while others show no significant association between widowhood and distress [13,16,40,41]. Some studies suggest that living with spouse could reduce the risk of anxiety [42], depression [43], loneliness [13], and other forms of psychological distress among left-behind parents [44]. This was also reflected in the qualitative data, where parents relied on spousal support to manage emotional distress. These findings underscore the importance of strengthening social support networks, especially for widowed parents, such as community-based programs and accessible mental health services, which could play an important role in improving their psychological well-being.

This study revealed that the parents with multi-morbidity had significantly higher odds of experiencing psychological distress (aOR: 4.168, 95% CI: 1.691–10.274, p = 0.022). This finding aligns with prior studies in Nepal [45], India [46], and China [43], which highlight a strong link between chronic disease and mental health deterioration. For instance, in China those left-behind parents with two or more chronic diseases experienced a higher prevalence of both depressive symptoms and moderate to severe depression compared to those with fewer chronic illnesses [43]. The qualitative narratives reinforced this association, as parents expressed concerns about managing their health conditions in the absence of their children, leading to heightened anxiety and stress. These findings suggest the need for home-based healthcare services and telemedicine initiatives to support aging populations with chronic conditions.

Parents whose children worked in labor-intensive jobs exhibited eightfold higher odds of psychological distress (aOR: 8.169; 95% CI: 1.351–49.403). The qualitative data further corroborated this finding by revealing concerns about harsh working conditions, job insecurity, and financial instability, which contributed to parental anxiety and worry. While limited research exists on migrant occupation and parental mental health, these findings highlight the need for stronger protections for labor migrants and improved communication between migrant workers and their families. The findings further emphasize the protective role of perceived social support. Parents who perceived low support from their migrant children had threefold higher odds of psychological distress (aOR: 3.797; 95% CI: 1.169–12.333). This is consistent with past studies highlighting the protective role of social support in mitigating mental health deterioration among parents and older adults [47,48]. The qualitative interviews further revealed that parents felt emotionally abandoned when their children failed to provide sufficient communication or financial support. Interestingly, while frequent communication is often considered a buffer against distress [43,49,50], this study found no significant association between distress and communication frequency, suggesting that the quality of interactions may be more crucial than their frequency.

This study is among the few that have assessed the prevalence of psychological distress and its associated factors among parents of emigrants, while also exploring their lived experiences. To enhance the reliability and depth of findings, efforts were made to minimize potential biases through structured data collection, pretesting of survey instruments, and the incorporation of both quantitative and qualitative methods. However, certain limitations should be acknowledged. Despite measures to reduce social desirability bias by ensuring participant confidentiality and conducting interviews in private settings, some respondents may still have underreported their distress or overemphasized positive coping mechanisms to align with cultural expectations. Additionally, recall bias might have influenced responses, particularly regarding past emotional experiences and changes in well-being over time. The study was conducted in urban areas of Nepal, where access to social networks, healthcare, and remittance benefits may not fully reflect the experiences of rural populations. While the findings remain highly relevant to migration-affected communities, variations in socioeconomic conditions, cultural expectations, and access to mental health resources should be considered when interpreting these results in different settings. Future research could benefit from incorporating rural populations and more diverse geographic contexts to capture a broader range of experiences. Another consideration is that, despite efforts to ensure balanced representation of male and female participants,

mothers were overrepresented in the sample due to their greater availability at home during data collection. Additionally, while financial support and frequency of communication were assessed, other psychosocial factors, such as the quality of communication, the nature of intergenerational conflicts, and parental expectations regarding children's return, were not explored in depth due to time and questionnaire constraints. Future research should explore the long-term mental health trajectories among left-behind parents, particularly as migration trends evolve and family structures shift.

## Conclusion

The study revealed that significant proportion of parents with emigrant children experience psychological distress, which is often influenced by health conditions, migrant employment status, and perceived social support. These findings highlight the need for targeted interventions to support left-behind parents, particularly those with chronic illnesses, limited social support, or children in labor-intensive jobs. Community-based mental health programs, social engagement initiatives, and strengthened intergenerational communication strategies could help mitigate distress.

## Supporting information

**S1 Table. Descriptive statistics of kessler psychological distress scale (K10) scores.**
(DOCX)

**S2 Table. Bivariate analysis of continuous K10 Scores across participant and migration-related characteristics.**
(DOCX)

**S1 File. Quantitative data.**
(XLSX)

**S1 Checklist. STROBE checklist.**
(DOCX)

**S2 Checklist. COREQ checklist.**
(DOCX)

## Acknowledgments

We share our gratitude to all the parents of the emigrant children who participated in this study and provided their valuable time and information. Without them, this study wouldn't have been possible.

## Author contributions

**Conceptualization:** Anjani Bhandari, Shishir Paudel, Anisha Chalise.

**Data curation:** Anjani Bhandari, Shishir Paudel.

**Formal analysis:** Anjani Bhandari, Shishir Paudel, Anisha Chalise.

**Methodology:** Anjani Bhandari, Shishir Paudel, Anisha Chalise.

**Supervision:** Shishir Paudel, Anisha Chalise.

**Validation:** Anjani Bhandari, Shishir Paudel, Anisha Chalise.

**Visualization:** Shishir Paudel, Anisha Chalise.

**Writing – original draft:** Anjani Bhandari, Shishir Paudel.

**Writing – review & editing:** Shishir Paudel, Anisha Chalise.

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
