## [Decision Letter · Decision Letter 0]

21 May 2025

PONE-D-25-12530Psychological Distress among Parents with Emigrant Offspring: A Mixed-Methods Study from Changunarayan Municipality, NepalPLOS ONE

Dear Dr. Chalise,

Thank you for submitting your manuscript to PLOS ONE. After careful consideration, we feel that it has merit but does not fully meet PLOS ONE’s publication criteria as it currently stands. Therefore, we invite you to submit a revised version of the manuscript that addresses the points raised during the review process.

The Reviewer has made important comments, and I invite you to address all of them.I especially encourage you to revise your Data Availability declaration. Saying that "all data are available without restriction" is too vague, and more clarity is needed as to how can readers access the data. In addition, given your research includes qualitatuve data, it's important to clarify if you have authorization from participants to share the original recordings or transcripts, or if there are confidentiality issues that would prevent you from sharing the qualitative data.

We look forward to receiving your revised manuscript.

Kind regards,

Ietza Bojorquez, Ph.D.

Academic Editor

PLOS ONE

Reviewers' comments:

Reviewer's Responses to Questions

**Comments to the Author**

1. Is the manuscript technically sound, and do the data support the conclusions?

Reviewer #1: Yes

2. Has the statistical analysis been performed appropriately and rigorously? 

Reviewer #1: Yes

3. Have the authors made all data underlying the findings in their manuscript fully available?

Reviewer #1: Yes

4. Is the manuscript presented in an intelligible fashion and written in standard English?

Reviewer #1: Yes

5. Review Comments to the Author

Reviewer #1: Summary and Overall Impression

This study used mixed methods to explore psychological distress among older parents whose children had emigrated from Nepal. The study identified factors associated with greater distress, including multiple morbidities, low support from emigrant children, etc. Overall the study explored an important and somewhat understudied topic. It was well done and well written.

Additional feedback provided in an attachment.

6. PLOS authors have the option to publish the peer review history of their article (what does this mean? ). If published, this will include your full peer review and any attached files.

**Do you want your identity to be public for this peer review?** For information about this choice, including consent withdrawal, please see our Privacy Policy .

Reviewer #1: No

---

## [Author Response · Author response to Decision Letter 1]

16 Jun 2025

Dear Editor,

We would like to thank you and the reviewer for your valuable time and comments provided for our manuscript entitled “Psychological Distress among Parents with Emigrant Offspring: A Mixed-Methods Study from Changunarayan Municipality, Nepal”. We have revised the manuscript in light of the comments provided and would like to provide a point-by-point response to all the provided comments here below:

Reviewer 1

Summary and Overall Impression: This study used mixed methods to explore psychological distress among older parents whose children had emigrated from Nepal. The study identified factors associated with greater distress, including multiple morbidities, low support from emigrant children, etc. Overall the study explored an important and somewhat understudied topic. It was well done and well written.

We thank you for your valuable time and kind words. We have revised our manuscript in reference to your valuable suggestion and are grateful for your support to improve our manuscript.

Major Issues

• Avoid use of “elderly” as it’s somewhat age-biased language. Also, 2.8% were aged 30-40 years and 34.9% were 40-50 years

o Thank you for your suggestion. We have omitted the use of term ‘elderly’ in the manuscript.

• It would be good to have more detail about the financial support from emigrant children. While not statistically significant (<.05) according to the chi-square test in Table 5, the p-value of .084 I think warrants exploring in the regression model. There may be possible interactions too that could be explored. It may be part of the reason that being employed as a laborer was associated with psychological distress.

o Thank you for your suggestion, we have added details about financial support in the interpretation of the result, as well as included it in the regression model.

• I wonder if the authors considered analyzing psychological distress as a continuous variable? There is some loss of data when recoding into categories.

o We thank the reviewer for the valuable suggestion regarding the potential information loss from categorizing psychological distress. In response, we conducted supplementary bivariate analyses using the continuous K10 scores and appropriate non-parametric tests (Mann–Whitney U and Kruskal–Wallis H). These analyses, presented in Supplementary Table S1, showed a pattern of associations largely consistent with those observed in the categorical analysis. While we acknowledge that categorization may reduce variability, in this case, the overall findings remained comparable. We have added these results to the supplementary material and briefly referenced them in the manuscript.

Minor Issues

• Age categories in Table 3 are overlapping

o Thank you for highlighting the error. We have revised the age category in the table ensuring it is not overlapping.

• In first paragraph of the discussion, the authors state “The qualitative findings revealed a complex emotional landscape, where some parents reported distress and loneliness, while others expressed pride in their children's achievements.” They should acknowledge that it’s possible for the same parents to have ambivalent feelings.

o Thank you for notifying us of this small yet critical oversight. The presence of ambivalent emotions among parents was indeed something we observed in our qualitative findings, but this was unfortunately lost in the original statement. We have now revised the sentence in the first paragraph of the discussion to accurately reflect this complexity: “The qualitative findings revealed a complex emotional landscape, where some parents reported distress and loneliness, while others expressed pride in their children's achievements and in some cases, the same parents experienced these ambivalent emotions simultaneously.”

• Several places where a space is needed before an in-text citation

o Thank you for notifying it. We have added space before in-text citations throughout the manuscript.

• Authors should clarify what diseases/conditions were considered

o Thank you for your helpful comment. In response, we have now provided a breakdown of the specific chronic conditions reported by participants in the Results section. This includes the most common conditions such as hypertension, diabetes, arthritis, asthma, chronic kidney disease, and other conditions like high cholesterol and cardiovascular diseases.

• On page 9 in the paragraph starting with “Parental well-being is deeply influenced by the migration of their children, often resulting in emotional and psychological challenges” The authors cite several studies of the prevalence of distress among left-behind parents, but I wonder if the rates are higher compared to those who do not have emigrant children? It seems some of the studies look at this, but more detail would be helpful about studies cited in the second half of the paragraph. This should also be clarified in the first paragraph of the discussion section.

o Thank you for your valuable suggestion. We have now incorporated the findings from other studies comparing the prevalence of mental distress among the parents with and without emigrant children, in the discussion section. And we have also highlighted that since our study does not explore this aspect and there are contradicting findings, future studies need to be conducted to explore this aspect in detail.

• The data have already been collected, but it would have been interesting to examine the potential relationship between emigrant children’s long-term plans (to return or not) and parents’ psychological distress. You note this in the limitations section but I wonder if you noticed any patterns in the qualitative data?

o Thank you for your valuable comment. We acknowledge that there might be potential relationship between emigrant children’s long-term plans (to return or not) and parents’ psychological distress. Though the plan of the children to return was not directly assessed as the parents were also not completely clear regarding their children’s plan, we have tried to assess it indirectly in both qualitative and quantitative phases. For instance, the nature of the visa of the children reflected that some of the children had permanent visa and residency, though it was not associated with psychological distress among the parents. In qualitative interviews, parents expressed thoughts such as:

Now he has no plan to return here. Instead, he requests us to move abroad with them. -IDI 7

I am quite hopeful they will return back and raise their kids with us. Me, and my wife can have our grandkids running around us soon. IDI 1.

My son and his family are living a kind of luxury life abroad, which our country lacks in a lot of ways. If I ask my son to leave everything and come back to us, that won’t be right. IDI 4

However, they didn’t express it in relation to their mental wellbeing.

• One of the review questions is “Have the authors made all data underlying the findings in their manuscript fully available?” If the authors have, they should include a statement about this in the methods section.

o Thank you for the comment. We have included a statement about data availability in the Data Availability Statement within the manuscript following journal guideline.

Editor Comment

• Thank you for your suggestion. We have revised our manuscript so as to meet the PLOS ONE requirements based on the templates provided.

• Thank you for your comment. In the ethical consideration subsection within the Method and materials section, we have specified that written informed consent was taken from all the participants prior to data collection. Our study did not include minors.

• Thank you for the comment. We have provided the captions accordingly.

• Thank you for your suggestion. We have reviewed the reference list ensuring that it is complete and correct.

I especially encourage you to revise your Data Availability declaration. Saying that "all data are available without restriction" is too vague, and more clarity is needed as to how can readers access the data. In addition, given your research includes qualitative data, it's important to clarify if you have authorization from participants to share the original recordings or transcripts, or if there are confidentiality issues that would prevent you from sharing the qualitative data.

• Thank you for your feedback. We have now revised the Data Availability Statement stating that the quantitative data is available as a supporting document and all relevant qualitative data are within the manuscript.

---

## [Editor Report · Decision Letter 1]

11 Jul 2025

Psychological Distress among Parents with Emigrant Offspring: A Mixed-Methods Study from Changunarayan Municipality, Nepal

PONE-D-25-12530R1

Dear Dr. Chalise,

We’re pleased to inform you that your manuscript has been judged scientifically suitable for publication and will be formally accepted for publication once it meets all outstanding technical requirements.

Kind regards,

Ietza Bojorquez, Ph.D.

Academic Editor

PLOS ONE